# Position: Accountable Deployment of Agentic AI Demands Layered, System-Level Interpretability

**Judy Zhu\*** [1]   **Dhari Gandhi\*** [1]   **Ahmad Rezaie Mianroodi** [2][1]   **Dhanesh Ramachandram** [1]   **Sedef Akinli Kocak** [1]
**Shaina Raza** [1]

\*These authors contributed equally to this work.

## Abstract

Agentic AI systems behave through trajectories: they plan, invoke tools, update memory, and coordinate over multiple steps. However, interpretability remains largely model-centric, focused on explaining single predictions rather than tracing long-horizon behavior and responsibility across interacting components. As a result, critical failures, such as tool misuse, coordination breakdowns, or goal drift, often evade existing audits until harm occurs. **We argue that interpretability for agentic systems must become system-centric, addressing trajectories, responsibility assignment, and lifecycle dynamics rather than internal model mechanisms alone.** We advance three claims: interpretability must (1) co-design with agentic capabilities, (2) address distinct layers of opacity with tailored methods, and (3) integrate across the deployment lifecycle. To operationalize this position, we introduce **ATLIS (Agentic Trajectory and Layered Interpretability Stack)**, a framework integrating five interpretability layers across a five-stage deployment lifecycle. ATLIS enables lightweight continuous monitoring with risk-aware escalation to deeper system-level analysis when incidents are detected. ATLIS provides a blueprint for closing the growing gap between agentic capabilities and the interpretability infrastructure needed to govern them. Project Page: https://vectorinstitute.github.io/ATLIS/.

**Position Statement**

The interpretability field is solving the wrong problem for the agentic era. Current methods explain how individual models compute outputs but cannot explain why an agent selected a particular plan, how multi-agent coordination failed, or where accountability lies within a system. We argue three points: (1) interpretability methods must co-design with agentic capabilities rather than follow them, embedding transparency into planning, tool use, and memory from the outset; (2) agentic opacity occurs at distinct layers such as behavioral, mechanistic, coordination, and safety, each requiring tailored methods; and (3) interpretability must integrate across the full agent development lifecycle rather than serve as a one-time audit.

## 1. Introduction

When an agentic system fails, who is accountable and how do we determine why? Unlike static models that produce isolated predictions, agentic systems plan, use tools, maintain memory, and coordinate actions over time (Abou Ali et al., 2025). Failures may arise not only from incorrect outputs, but also from evolving internal state and errors across components that are difficult to trace. Auditing agentic systems thus requires attention to both internal representations and system-level information flow, including how errors are introduced and propagated, to support accountability and safe deployment (Farooq et al., 2025). This presents fundamental challenges for traditional model-centric interpretability methods that are focused on single-step predictions (Murugesan, 2025).

Recent deployments illustrate that unsafe outcomes stem not from isolated model errors but from cumulative system-level dynamics. A preliminary evaluation by the National Highway Traffic Safety Administration (National Highway Traffic Safety Administration, 2025) found that semi-autonomous driving incidents involved failures spanning perception, planning, and human-machine handoff, including missed traffic signals, unexpected lane changes, and inadequate driver warnings, none of which were attributable to a single model output. Similarly, an investigation by The Markup (The Markup, 2024) revealed that a New York

---

[1]Vector Institute for Artificial Intelligence, Ontario, Canada. [2]Dalhousie University, Nova Scotia, Canada. Correspondence to: Shaina Raza <shaina.raza@vectorinstitute.ai>, Judy Zhu <judy.zhu@vectorinstitute.ai>.

*Proceedings of the $43^{rd}$ International Conference on Machine Learning*, Seoul, South Korea. PMLR 306, 2026. Copyright 2026 by the author(s).

City municipal chatbot advised employers they could take workers' tips and that landlords could discriminate against voucher recipients, violating local law. These failures arose from inadequate constraint enforcement, poor grounding to authoritative sources, and absent lifecycle oversight.

Model-oriented interpretability techniques, such as feature attributions or reasoning traces, are designed to explain individual model outputs, not the behavior of systems composed of multiple agents acting over time (ŞAHiN et al., 2024). Even when individual agents perform correctly in isolation, system-level opacity can arise from the interaction of planning, memory, and delegation mechanisms. For example, model-centric interpretability might explain why a language model selected a particular token or generated a specific response using feature attributions or reasoning traces. In an agentic system, agent refers to LLM-powered entities that plan, use tools, and communicate via natural language (e.g., AutoGen, CrewAI, LangGraph-style architectures). The final decision may depend on how multiple components interact across time. Therefore, while model-level analysis may provide insight into individual steps, the overall behavior emerges from the interaction of memory updates, tool use, and coordination across components. Examining single model outputs in isolation would not be able to completely explain this. This leaves a critical gap for interpretability approaches that reflect how agentic systems operate in practice, rather than how individual models behave in isolation.

In this study, we define *agentic system interpretability* as the ability to explain, trace, and audit how decisions arise within an agentic system, including how reasoning, memory, tools, and coordination across agents contribute to outcomes. Current interpretability practices remain largely *model-centric*, focused on explaining individual model outputs through techniques such as feature attributions, saliency maps, and circuit analysis (Gao & Guan, 2023). These methods do not capture how decisions emerge from interactions across multiple components over time. Therefore, **we argue that interpretability for agentic systems must move from model-centric analysis to system-centric layered analysis**, where system-centric implies tracing behavior, responsibility, and information flow across the full agentic architecture.

Recent work addresses specific aspects of agentic interpretability: automated failure attribution for identifying which agent failed (Zhang et al., 2025), traceability in linear pipelines (Barrak, 2025), and causal metrics for collective behavior (Itzhak Weinberg, 2025). However, these methods target individual diagnostic tasks rather than the architectural shift required to govern agentic systems end-to-end. Our position differs by defining the necessary layers of agentic system observability and integrating them across the deployment lifecycle. Rather than post-hoc blame assignment, we offer a blueprint that situates existing diagnostic tools within a layered framework and identifies where current methods fall short.

## 2. Our Position

Interpretability research is becoming misaligned with what matters in the agentic era. Most current interpretability methods explain isolated aspects of predictive models rather than end-to-end systems (Subhash et al., 2022). *Glass-box* approaches, where interpretability is built in by design, have been more confined to simpler model classes and do not readily extend to the multi-component pipelines typical of agentic deployments (Liu et al., 2025b). Meanwhile, *post-hoc* techniques such as SHAP and Integrated Gradients remain primarily model-centric, characterizing input-output behavior via feature attributions, but offering limited visibility into system-level reasoning, tool use, memory updates, and decision propagation across time (Salih et al., 2024). Surrogate approaches such as LIME approximate complex behavior with simpler interpretable models (Tan et al., 2023). Mechanistic interpretability methods, including circuit analysis and feature discovery, reverse-engineer internal components to explain how models produce outputs (Bereska & Gavves, 2024). However, as systems grow more complex, behavior cannot be reduced to isolated model mechanisms.

Recent interpretability work in industry and frontier labs, particularly at organizations such as OpenAI (OpenAI et al., 2023), Anthropic (Anthropic, 2024), and DeepMind (Google DeepMind, 2025), has made substantial progress on model-level analysis in frontier language models. This includes sparse feature discovery (OpenAI, 2024; Gross et al., 2024), circuit-level tracing (Ameisen et al., 2025), and biologically inspired representations of internal structure (Lindsey et al., 2025). These contributions focus on how a single model generates token-level outputs, while agent-level phenomena such as plan selection, tool routing, memory updates, and coordination failures across interacting components remain underexplored. In parallel, *agentic system interpretability* shifts the focus of explanation from static model internals to trajectories of actions, state, and interaction (Kim et al., 2025).

Empirical studies of multi-agent failures also show that many breakdowns arise from cross-component interactions and cross-step dynamics, rather than from any single model decision (Pan et al., 2025). As a result, the core challenge is not only interpreting individual models, but achieving observability and governance of agentic systems as a whole. This requires standardized logging of actions, tool use, and intermediate states across large language models (LLMs) and agent orchestration frameworks.

These gaps motivate our position that **interpretability for the agentic era must expand from model-centric expla-**

nations to system-level analysis of behavior, interaction, and accountability. To support this, we advance three claims:

- **Claim 1: Co-design over reaction.** Interpretability methods, at a research level, must co-develop alongside agentic capabilities. In this paper, we use co-design to imply intentional, simultaneous, mutually-informing development of two distinct subsystems, capabilities and interpretability infrastructure, rather than interpretability being added retrospectively after deployment. Co-design does not imply continual redesign of the full interpretability stack. Rather, it reflects iterative refinement of architecture-sensitive interpretability layers while preserving broader system-level monitoring, governance, and accountability mechanisms.

- **Claim 2: Layered decomposition.** Agentic opacity occurs at distinct layers (e.g., behavioral patterns, internal circuits, abstraction mappings, multi-agent coordination, and safety constraints). Transparency requires tailored and layered analysis.

- **Claim 3: Lifecycle integration.** Interpretability must be applied throughout the full agent deployment lifecycle, rather than as a one-time audit.

To action these claims, we introduce **A**gentic **T**rajectory and **L**ayered **I**nterpretability **S**tack (**ATLIS**) (Section 4), a five-layer framework that breaks down agentic system transparency requirements and maps them to system development stages.

## 3. Alternative Views

Our position is not without challenges. We address four primary objections below.

**Alternate View 1: Agentic interpretability should follow, not be designed in parallel.** A common view is that interpretability for agentic systems should come after mature architectures are developed, rather than changing alongside them. Investing too early may not help, since future agents could rely on very different representations or planning methods than today's systems. For example, methods based on natural-language reasoning traces, such as chain-of-thought explanations (Barez et al., 2025), assume that agents can reason in a way that reliably reflects their internal decision processes. This view invokes *Polanyi's Paradox*, where behaviour can be competent without being able to explicitly explain how they do so (Autor, 2014). Under this analogy, the behavior of agentic systems can be shaped and constrained through training, feedback, and incentives, even if their internal decision-making processes remain opaque (Ouyang et al., 2022).

**Response:** Interpretability research has historically lagged behind capability development, leaving practitioners dependent on explanations that are difficult to validate and misaligned with how systems actually behave (Xua & Guanci, 2024). Retrofitting interpretability after deployment is harder than building it in from the start. Agentic systems compound this risk: hidden goals, internal conflicts, or deceptive strategies may remain undetected during development and surface only in deployment (Ngo et al., 2024). Co-designed agentic interpretability ensures that transparency mechanisms are tested and refined alongside capabilities, rather than bolted on after failures occur.

**Alternate View 2: Behavioral control is sufficient.** A second objection holds that detailed interpretability is not required for safe deployment. Alignment can be achieved through behavioral steering mechanisms such as reinforcement learning from human feedback (RLHF) and constitutional objectives, combined with rigorous black-box evaluation (Bai et al., 2022; Ouyang et al., 2022). From this perspective, strong performance on benchmarks, robustness tests, and behavioral checks are sufficient to build trust, making system-level interpretability optional.

**Response:** Behavioral testing may show that a system performs well on average, but cannot explain why a multi-step plan failed or how errors propagated across components. Without system-level interpretability, failures cannot be reliably traced, audited, or corrected, even when alignment methods appear effective in standard evaluations (Rudin, 2019; Sengupta et al., 2025; Mallen et al., 2024). Agentic systems amplify this gap: failures may emerge from interactions among planning, memory, and tool use that black-box testing does not probe. Interpretability is not opposed to behavioral control but complementary to it.

**Alternate View 3: Holistic rather than layered interpretability.** A third critique challenges our layered decomposition proposed in our framework (Section 4). This view argues that agentic systems should be explained as integrated wholes rather than decomposed into separate layers. Separating behavioral, mechanistic, coordination, and safety layers may obscure tightly coupled interactions and produce fragmented explanations that miss cross-layer dependencies (Raza et al., 2024). Unified interpretability frameworks that synthesize explanation properties across the system (Subhash et al., 2022) or map latent representations directly to interventions (Bhalla et al., 2025) may better capture emergent behavior.

**Response:** Layered interpretability does not treat layers as isolated silos. Rather, it decomposes the system into analytically distinct levels while preserving the ability to trace interactions across them. Empirical studies show that failures often originate in specific subsystems, such as planning or coordination, even when they propagate system-wide (Pan

et al., 2025). Distinguishing a tool-routing error from a planning failure requires layer-specific analysis before tracing how the error cascaded. Treating the system as a single undifferentiated unit is often too coarse for debugging, mitigation, and governance.

**Alternate View 4: Agentic lifecycle-integrated interpretability is computationally impractical.** A fourth objection accepts the value of interpretability in principle but argues whether continuous integration across the deployment lifecycle is feasible. Runtime logging, monitoring, and tracing can add computational overhead, increase latency and complicate deployment creating challenges in multi-agent systems operating under real-time constraints (Zaharia et al., 2024; Pan et al., 2025). From this view, interpretability should be limited to pre-deployment audits or activated only after failures occur.

**Response:** Practical constraints are real, but restricting interpretability to pre-deployment audits or post-failure analysis is insufficient for governing adaptive agentic systems. Agent behavior evolves through tool use, memory updates, and interaction with dynamic environments; problems that were absent at deployment can emerge at runtime. ATLIS features integrated interpretability layers (Section 4.2) that address this through lightweight, risk-aware monitoring: Layer 1 behavioral tracking runs continuously with low overhead, while deeper analysis (Layers 2–4) activates only when anomalies are detected. This graduated approach balances interpretability with computational cost.

## 4. Our Proposal: ATLIS Framework

To operationalize the position advanced in Section 2, we introduce the **Agentic Trajectory and Layered Interpretability Stack (ATLIS)**. The framework addresses each of our three claims: *lifecycle integration* (Claim 3) is reflected via five deployment stages that specify *when* interpretability must be applied; *layered decomposition* (Claim 2) is ensured through five interpretability layers that specify *what* analysis is required; and *co-design over reaction* (Claim 1) is supported by embedding interpretability requirements into each phase from pre-deployment onward, ensuring that transparency mechanisms develop alongside agent capabilities rather than being retrofitted after failures occur. Basic definitions that distinguish agentic interpretability from model-centric approaches are provided in Appendix A.

### 4.1. Agentic System Deployment Lifecycle Stages

This section operationalizes our first claim (*co-design over reaction*) and third claim (*lifecycle integration*). Post-hoc interpretability is insufficient because: (1) runtime anomaly detection requires pre-deployment baselines; (2) incident response depends on structured logs, even with full logs,

automated attribution achieves only 53% agent-level and 14% step-level accuracy (Zhang et al., 2025); and (3) governance evolves through iterative feedback loops rather than static controls. We therefore organize deployment around five stages, each linked to the interpretability layers defined in Figure 1.

**Stage 1: Pre-Deployment.** The input to this stage is requirements and risk assessment. This stage covers problem framing, system specification, architectural design, and risk assessment before agents are exposed to real-world environments. Developers define goals, task decomposition, memory structures, tool interfaces, and evaluation strategies (Yuksel et al., 2025). Simulation, offline benchmarking, and adversarial testing are used to identify likely failure modes with downstream impacts prior to deployment. The output is an intepretability plan for what will be monitored and what "normal" system behavior looks like.

**Stage 2: Deployment.** The input to this stage is interpretability instrumentation plan. This stage activates behavioral logging and monitoring, including action sequences, tool usage protocols, and latency profiles, to define normal system operation in production launch. It also initializes sampling and probing hooks for deeper analysis when needed, begins coordination monitoring for multi-agent settings, and enables graduated safety interventions (e.g., human-in-the-loop escalation). The output is an operational dashboard with anomaly alerts.

**Stage 3: Post-Deployment Runtime Operations.** The input to this stage is the operational dashboard and behavioral baselines. During runtime, agent actions are tracked in real time to detect anomalies, behavioral drift, and deviations from expected trajectories. This stage performs continuous Layer 1 drift detection, periodic Layer 3 reasoning checks, Layer 4 coordination health monitoring, and Layer 5 constraint validation (Li et al., 2024; He et al., 2025). The output is anomaly and drift alerts with cross-layer flags to support tracing.

**Stage 4: Incident Response.** The input to this stage is anomaly flags and structured logs. This stage is triggered when agents behave in unsafe, incorrect, or misaligned ways. Incidents may arise from planning errors, tool misuse, conflicting internal states, or coordination failures across agents. This stage involves full cross-layer tracing following minimal sufficient depth (Raza et al., 2025; Liu et al., 2025a). The output is an an incident report with root-cause attribution and recommended mitigations.

**Stage 5: Post-incident Learning.** The input to this stage is an incident with root-cause attribution and recommended mitigations. Incidents are treated as structured evidence for improving system robustness, evaluation, and governance continuously. Logs, traces, and failure cases are tracked and

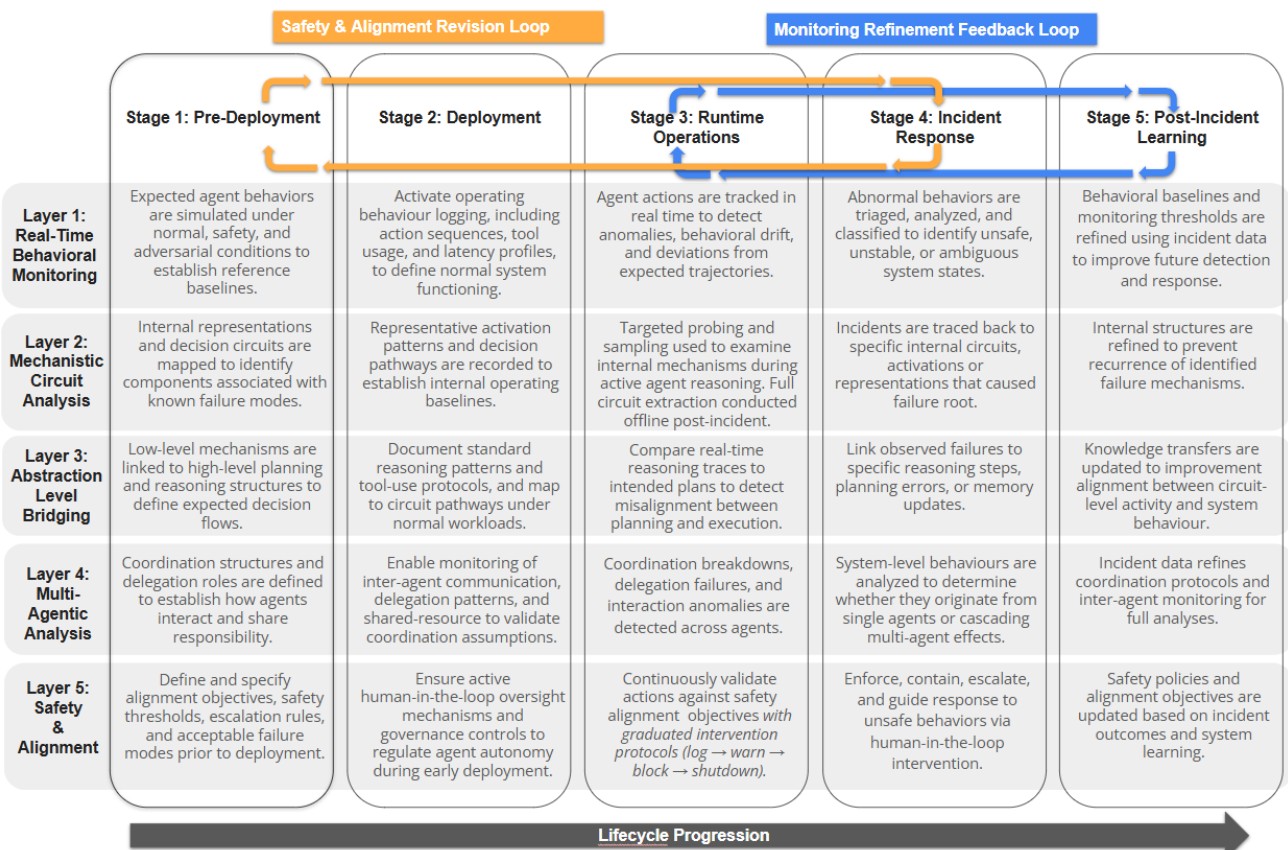

*Figure 1.* ATLIS (Agentic Trajectory & Layered Interpretability Stack) is a deployment lifecycle and integrated interpretability stack for agentic systems. This framework integrates five interpretability layers across the agentic system lifecycle: (1) Real-Time Behavioral Monitoring tracks observable agent actions; (2) Mechanistic Circuit Analysis examines internal model representations; (3) Abstraction-Level Bridging connects low-level circuits to high-level reasoning; (4) Multi-Agent Analysis evaluates coordination dynamics; and (5) Safety and Alignment ensures adherence to predefined objectives. The framework incorporates two loops: blue arrows denote the monitoring refinement feedback loop, while orange arrows denote the safety and alignment revision loop. Computational overhead ranges from low (Layer 1 continuous monitoring) to high (Layer 2 full circuit extraction during incident response). ATLIS also supports comparison across repeated execution trajectories to identify behavioral drift and recurring coordination failures over time.

used to update test suites, safety constraints, and control policies (Cruz, 2025). The output is updated baselines, detectors, and instrumentation that feed back into runtime operations and future deployments.

Two feedback loops enable co-design (Figure 1). In practice, escalation across feedback loops depends on the persistence, severity, and propagation of observed anomalies. Recurring unresolved deviations may trigger monitoring refinement, while failures involving safety violations, cascading coordination breakdowns, or cross-layer propagation beyond local containment may activate safety and alignment revision processes. The **Monitoring Refinement Feedback Loop** (Stages 3→4→5→3) uses runtime drift signals and incident outcomes to recalibrate detectors and monitoring thresholds. The **Safety & Alignment Revision Loop** (Stages 1→2→3→1) updates pre-deployment assumptions, tests, and safety objectives based on deployment and run-

time observations; severe incidents additionally trigger expedited revisions that propagate from Stage 4 back to Stage 1.

Layers differ in their dependence on specific model architectures. Lower architectural dependence indicates that interpretability mechanisms operate primarily on observable behaviors, workflows, or governance signals, whereas higher dependence reflects tighter coupling to internal representations and current architectures (e.g., transformer-based planning, chain-of-thought reasoning). ATLIS' limited architecture-specific dependencies allow for stable governance through architecture changes, preventing the need for re-engineering across all framework layers, further supporting co-design.

### 4.2. Layered Interpretability Stack

This section operationalizes our second claim (*layered decomposition*): agentic opacity occurs at distinct layers, each

*Table 1.* Layered Interpretability Stack: Methods, Computational Overhead, and Outputs

| Layer | Representative and Emerging Methods | Overhead | Outputs | Architectural Dependence |
|---|---|---|---|---|
| L1: Real-Time Behavioral Monitoring | Action entropy, tool-call frequency analysis, isolation forests, AD-WIN drift detection | Low | Anomaly flags, behavioral logs, drift alerts | Low |
| L2: Mechanistic Circuit Analysis | Sparse autoencoders (Huben et al., 2023), probing classifiers, circuit discovery (Conmy et al., 2023) | High | Circuit diagrams, feature attributions, causal graphs | High |
| L3: Abstraction-Level Bridging | Causal tracing (Meng et al., 2022), CoT alignment verification (Barez et al., 2025), logit lens (Nanda et al., 2023) | Moderate | Reasoning traces, abstraction mappings, alignment scores | Moderate |
| L4: Multi-Agent Analysis | Delegation graph tracing, shared-memory logging, interaction graph anomaly detection | Moderate–High | Coordination graphs, influence maps (Parfenova et al., 2025), bottleneck reports | Low-Moderate |
| L5: Safety and Alignment | Constraint satisfaction monitoring, reward hacking detection (Ngo et al., 2024), goal drift measurement, alignment probing (Burns et al., 2024), red-teaming | Variable | Compliance reports, alignment scores, risk assessments | Low |

requiring tailored analysis. Model-centric interpretability treats the system as a single unit; we argue instead for layered decomposition that distinguishes behavioral patterns, internal circuits, abstraction mappings, multi-agent coordination, and safety constraints. This decomposition supports targeted diagnosis: failures may originate at any layer in practice, and meaningful explanations often require linking evidence across different layers. Each layer in ATLIS corresponds to a different scope of explanation and responsibility. Layer 1 focuses on observable behavioral monitoring (what the system is doing and whether behavior deviates from expectations). Layers 2 and 3 examine intra-agent reasoning and internal mechanisms (how individual modules reason, plan, retrieve memory, and produce decisions). Layer 4 analyzes inter-agent coordination and execution dynamics across workflows (how interactions between agents produce system-level outcomes). Layer 5 addresses governance, oversight, and safety constraints across the deployment lifecycle (whether the overall system remains aligned with operational and policy requirements). While analytically distinct, these layers are intended to operate together, allowing evidence to be traced across levels during incident analysis. We embed this integrated interpretability stack within the agentic lifecycle, as illustrated in Figure 1 and described in Table 1.

**Layer 1: Real-Time Behavioral Monitoring.** The input to this layer is structured behavioral telemetry, including agent action sequences, tool-call traces, latency and resource profiles, and state-transition logs. This layer captures surface-level deviations as behavior unfolds in real time. This monitoring may also aggregate behavioral patterns across repeated execution trajectories to detect drift, recurring coordination failures, or systematic deviations that are not observable within a single runtime episode. Representative techniques include action-sequence entropy to detect irregular trajectories (Okpala et al., 2025), tool-call frequency analysis for unexpected resource usage, temporal anomaly detection (e.g., isolation forests) over behavioral embeddings, and drift detectors such as ADWIN or Page–

Hinkley to flag distributional shifts over time (Pinto et al., 2019). While Layer 1 provides high recall for identifying *what* the agent did, it cannot explain *why* the behavior emerged, motivating deeper analysis. The output of this layer is a continuous stream of behavioral baselines, anomaly flags, drift alerts, and logging artifacts that support downstream tracing.

**Layer 2: Mechanistic Circuit Analysis.** The input to this layer is internal model evidence, including hidden activations, attention patterns, and targeted probe signals collected from planning, memory retrieval, and tool-selection steps. Layer 2 focuses on causal mechanisms inside the agent that shape downstream actions. Core methods include sparse autoencoders (SAEs) for discovering interpretable features in residual streams (Huben et al., 2023), and probing classifiers for concept decodability, and circuit discovery for localizing failure roots (Conmy et al., 2023). Because full circuit extraction is computationally expensive, this layer is typically activated selectively during validation or incident response. The output of this layer is circuit-level attributions, causal graphs, and mechanistic explanations that identify which internal structures produced the observed behavior.

**Layer 3: Abstraction-Level Bridging.** The input to this layer is intermediate reasoning evidence, including high-level plans, step-wise traces, memory updates, and abstract decision representations. Layer 3 bridges the semantic gap between low-level circuits and reasoning, which is critical because multi-step planning errors may only manifest after several actions. There is some emerging work that suggests bridging between mechanistic model-level interpretability with agentic system-level interpretability. Diagnosing failures in LLM-based multi-agent systems can benefit from layered summaries of what agents were trying to do, what actions they took, and what execution steps actually occurred (Sheng et al., 2026); and mechanistic signals, detected through sparse auto-encoders, and higher-level LLM behavioural summaries can serve as complementary channels for interpreting LLM agent behaviour (Yan

et al., 2026). The output of this layer is an abstraction mapping that links internal mechanisms to specific reasoning steps, decision points, or memory operations, yielding interpretable trajectory-level explanations.

**Layer 4: Multi-Agentic Analysis.** The input to this layer is interaction-level evidence from multi-agent deployments, including inter-agent communication transcripts, delegation and handoff graphs, shared-memory access logs, and co-ordination signals. This layer addresses the complexity of systems composed of multiple coordinating agents, where failures are often emergent rather than attributable to any single component. Recent research demonstrates that analyzing these multi-agent coordination dynamics provides essential visibility into emergent system behaviors, effectively complementing model-internal interpretability methods (Parfenova et al., 2025). While computational overhead scales with the number of agents and interaction frequency, lightweight sampling methods can be employed to manage costs in large ensembles. The output of this layer is a suite of coordination diagnostics, influence maps, and interaction anomaly reports that explain failures at the holistic system level.

**Layer 5: Safety and Alignment.** The input to this layer is governance-relevant evidence, including safety objectives, policy constraints, escalation thresholds, alignment probes, and red-teaming results. Layer 5 evaluates whether agent behavior remains within acceptable operational and ethical boundaries, situating insights from lower layers within oversight requirements. Representative methods include satisfaction monitoring (verifying adherence to hard-coded rules and soft policy preferences), reward hacking detection (Ngo et al., 2024) alignment probing under distribution shift (Burns et al., 2024), and adversarial stress testing pipelines (Ganguli et al., 2022). The output of this layer is compliance and risk assessments, intervention triggers, and updated safety policies that connect interpretability artifacts to accountability and governance.

Together, these layers form a structured interpretability stack tailored to agentic systems. The lifecycle stages clarify *when* interpretability is required; the layered stack clarifies *what kind* of interpretability is needed and *where* failures may arise. By integrating both dimensions, ATLIS enables system-level interpretability as advocated in our position.

### 4.3. Example: Palliative Care Referral

To ground ATLIS in practice, consider a hospital deploying an agentic system design to support palliative care referral for patients with treatment-resistant cancer. The hospital runs two separate systems each in different disease sites: lung and gastrointestinal (GI). Each site uses the same agentic referral workflow, with recommendations routed to clinicians for approval and audit logging. Figure 2 illustrates

this diagnostic pathway, showing how ATLIS layers activate across the lifecycle to detect, trace, and resolve the referral timing divergence.

Despite identical referral criteria and subagent architectural components, the lung site deployment begins referring patients later than the GI site deployment for matched risk profiles. In one system, short-term symptoms improvement stored in memory delays escalation; in the other, accumulated hospitalizations trigger earlier referral. The divergence emerges gradually across repeated referral episodes due to differences in how longitudinal evidence is stored, weighted, and propagated through monitoring and planning.

**How ATLIS Surfaces the Divergence:** ATLIS surfaces this drift before it causes harm by identifying systematic deviations across repeated execution trajectories of the deployed referral workflow, flagging delayed lung referrals relative to GI referrals for comparable patient profiles.

It then traces the root cause through Layers 2-4 using inter-agent coordination signals and mechanistic differences. This pinpoints the divergence to differences in memory weighting and inter-agent handoff dynamics. Layer 3 maps these differences in symptom versus hospitalization weighting to divergent clinical reasoning pathways, while Layer 5 evaluates whether referral timing remains within prescribed safety bounds and escalates borderline cases for human-in-the-loop clinician review. Because ATLIS is embedded across the deployment lifecycle, these findings feed back into updated simulations (pre-deployment), refined runtime baselines (operations), and recalibrated orchestration policies (post-incident learning), enabling continuous system correction rather than one-time diagnosis.

**Why Model-Centric Methods Would Fail Here:** Model-centric interpretability would likely miss this failure. Feature attribution methods (e.g., SHAP or Integrated Gradients) explain individual predictions, but the divergence arises from longitudinal interactions between monitoring, memory, planning, and referral timing. Chain-of-thought inspection or single-agent circuit analysis may appear locally consistent, while the true cause lies upstream in memory updates and cross-agent handoff dynamics. Without behavioral drift baselines and coordination-level analysis, the system-level mechanism would remain hidden.

## 5. Call to Action and Implementation Plan

Adopting system-centric interpretability for agentic systems raises several priority research directions and has broader implications requiring concerted effort from interdisciplinary groups.

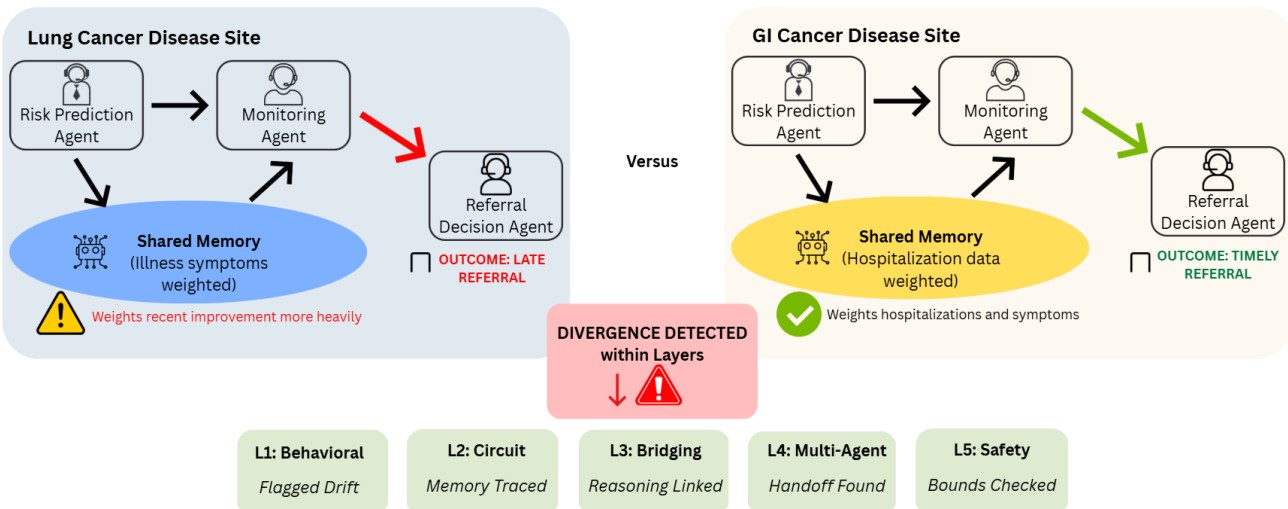

*Figure 2.* This is an example of implementing ATLIS for healthcare referrals. Two agentic systems, with identical architectures and referral criteria, begin producing different referral timing for patients with comparable risk profiles. This divergence emerges from how accumulated hospitalizations trigger earlier referral in one system.

## 5.1. Research Directions

**System-Level Attribution.** Current interpretability methods attribute outputs to model internals, but agentic outcomes emerge from multi-step trajectories spanning planning, memory, tool use, and delegation. New attribution methods are needed that track how decisions propagate across components over time (Xia et al., 2025), including formal causal frameworks for responsibility assignment in multi-component systems and techniques for reconstructing decision paths under partial observability.

**Scalable Runtime Monitoring.** Mechanistic analysis remains computationally prohibitive for continuous deployment. Lightweight alternatives are needed, such as multi-resolution observability that logs coarse artifacts continuously while reserving expensive analysis for flagged episodes (Dong et al., 2024), minimal sufficient statistics for drift detection (Rath, 2026), and selective activation strategies that trigger deeper interpretability only when risk thresholds are exceeded.

**Benchmarks and Evaluation.** There is a lack of standardized resources for evaluating system-centric interpretability. Thus, progress requires shared benchmarks capturing agentic phenomena (coordination failures, goal drift, emergent behaviors), common logging schemas, trajectory datasets with ground-truth failure annotations, and evaluation protocols for multi-agent settings.

## 5.2. Broader Implications

Our position is strengthened by interdisciplinary developments and extends beyond the research community into engineering, policy and governance. While *researchers*

need to formalize this stack as a modular research object to test methods across orchestration and memory layers, publish reproducible reference architectures, and log standards, *practitioners* should build traceability instrumentation at each module boundaries to clarify why specific plans or tool actions occur. This supports guardrail implementation around tool use under uncertainty. Beyond technical implementation, *regulators* must shift compliance standards to require system-level evidence, such as mandating tracing requirements, full lifecycle monitoring plans, and decision provenance as part of approval or procurement processes, rather than relying solely on model performance metrics. Encouraging shared learning mechanisms, such as standardized incident taxonomies, can encourage proactive interpretability integraton within deployed agentic systems and earlier disclosures of failures. *Organizations* deploying agentic systems should perceive agentic systems as continuously governed entities, adopting layered auditing and incident response loops as core acceptance criteria.

Closing this gap requires a coordinated effort to standardize reporting mechanisms and organizational change management practices that accommodate the non-linear complexity of multi-agent systems. Governance frameworks may require system-level evidence rather than relying solely on model performance metrics. These applications, however, depend on first establishing the technical foundations outlined above.

## 5.3. Scope and Assumptions

Our position rests on some assumptions that warrant acknowledgment. First, we assume that interpretability is necessary for accountability, an assumption that behavioral-

control proponents would dispute (Alternate View 2). Second, ATLIS provides a conceptual framework rather than a complete adoption plan. Third, as agentic systems grow in autonomy and interaction complexity, tracing internal states across planning, memory, tool use, and multi-agent communication becomes increasingly difficult. Our framework does not assume that full internal transparency is always achievable; rather, it supports **selective**, **layered**, and **risk-aware** interpretability calibrated to deployment criticality and acceptable overhead. Developing scalable methods for tracking agentic interactions remains an open challenge.

## 6. Conclusion

Agentic systems mark a shift from models that predict to systems that act, yet interpretability research has not kept pace with this transition. This paper argues that interpretability for agentic systems must be system-centric rather than model-centric, formalized through three claims: interpretability must coevolve with agentic capabilities, it must address multiple layers of system behavior, and it must operate across the full agent lifecycle. To make this position actionable, we introduce ATLIS, a framework which structures interpretability around behavioral, mechanistic, abstraction, coordination, and safety layers aligned with lifecycle stages from deployment through post-incident learning. As autonomous systems become embedded in high-stakes domains, system-centric interpretability will be essential not only for understanding how agents behave, but for ensuring that they can be trusted to act safely and reliably.

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

# Appendix

## A. Formal Formulation of ATLIS

**Definition A.1** (Agentic Trajectory). An agentic trajectory is a sequence $\tau = (s_0, a_0, s_1, a_1, \ldots, s_T)$ where each state $s_t \in \mathcal{S}$ captures the full system configuration, including agent memory, environment state, tool outputs, and inter-agent message history, and each action $a_t \in \mathcal{A}$ is executed by an active agent $\phi(t) \in \mathcal{N}$, where $\mathcal{N} = \{1, 2, \ldots, N\}$ is the set of agents and $\phi : \mathbb{N} \to \mathcal{N}$ is the turn-selection function.

This definition emphasizes that agentic systems produce *trajectories*, not isolated outputs. The state $s_t$ is not merely the input to a model but the cumulative result of prior actions, memory updates, and environmental feedback. Interpretability methods designed for input-output mappings cannot capture this temporal structure.

**Definition A.2** (System Outcome). For a trajectory $\tau$, the outcome function $Z : \mathcal{T} \to \{0, 1\}$ indicates task failure ($Z(\tau) = 1$) or success ($Z(\tau) = 0$). An outcome may depend on any subset of states and actions in $\tau$, not only the final state $s_T$.

**Definition A.3** (Layered Explanation). A layered explanation for outcome $Z(\tau)$ is a tuple $E = (E_1, E_2, E_3, E_4, E_5)$ corresponding to the five layers of the ATLIS stack. $E_1$ captures behavioral evidence: observable action patterns, tool invocations, and state transitions. $E_2$ captures mechanistic evidence: internal representations, decision circuits, and activation patterns within individual agents. $E_3$ captures abstraction mappings that link low-level mechanisms to high-level reasoning steps. $E_4$ captures coordination dynamics: delegation structures, information flow, and inter-agent dependencies. $E_5$ captures safety and alignment status: constraint satisfaction, alignment metrics, and governance violations. Each component $E_k$ may be empty if Layer $k$ analysis is not required for the explanation at hand.

**Definition A.4** (Attribution Completeness). An explanation $E$ is *complete* for a failure at time $t^*$ if there exists a traversal path $P \subseteq \{1, 2, 3, 4, 5\}$ such that the union $\bigcup_{k \in P} E_k$ jointly identifies the responsible agent $i^* = \phi(t^*)$, establishes a causal chain linking action $a_{t^*}$ to outcome $Z(\tau) = 1$, and can be computed within the resource constraints of the deployment context.

The third condition i.e., computational feasibility acknowledges practical limits. A complete explanation must be *achievable*, not merely conceivable. Full mechanistic circuit extraction may be justified during incident response but prohibitive during continuous runtime monitoring. This motivates our lifecycle integration, which specifies when computationally expensive analyses are warranted versus when lightweight monitoring suffices.

**Definition A.5** (Minimal Sufficient Depth). A traversal path $P$ has *minimal sufficient depth* if removing any layer $k \in P$ would render the explanation incomplete.

This principle guides efficient interpretability: begin at the layer where symptoms manifest, traverse only to layers required for causal attribution, and terminate when the root cause is solvable. A behavioral anomaly (Layer 1) that traces to a tool-routing error need not invoke full circuit analysis (Layer 2) if the delegation graph (Layer 4) suffices to identify the misconfigured handoff.

Together, these definitions formalize what it means to explain agentic system behavior. Unlike model-centric interpretability, which attributes outputs to input features or internal neurons, system-centric interpretability must attribute *outcomes* to *trajectories* through *layered* analysis. The trajectory captures temporal dynamics; the layers capture distinct sources of opacity; the traversal path captures the reasoning needed to connect symptoms to causes.

