# OpenReview forum: "Position: Accountable Deployment of Agentic AI Demands Layered, System-Level Interpretability"
_ICML.cc/2026/Position_Paper_Track — ICML 2026 Position Paper Track regular_

### Official Review · Reviewer_e69w · 2026-03-04

**Significance:** 3
**Argument Clarity:** 4
**Rating:** 5
**Confidence:** 4

**Questions:**

Is something missing in ATLIS to explicitly consider comparative behaviour across system instances?

**Alternative Views Section:**

Yes

**Compliance With Llm Reviewing Policy A Conservative:**

Affirmed.

**Discussion Potential:**

3

**Final Justification:**

The rebuttal addressed my main concerns. My original accept evaluation was reinforced.

**Paper Summary:**

This paper takes the position that explainability/accountability  of agentic systems requires system-level, proactive rather than posthoc, integration into the stack of so agent reasoning technologies, and that it should be done in a layered systematic manner that follows a certain scope of responsibility at each layer.  To provide guidelines to this it presents the Agentic Trajectory and Layered Interpretability Stack (ATLIS) that presents down agentic system transparency requirements and maps them to system development stages.

**Position:**

Yes

**Position In Title:**

Yes

**Related Work:**

3

**Strengths And Weaknesses:**

# Strengths

The position paper is well structured and clearly laid out. It presents both its position and possible alternative views consistently and without obvious bias. The position it takes is definite enough to be a clear position that can attacked and defended, without being so detailed that it is a fully developed proposed approach itself. I am not an expert in this area, so I can’t tell if the citations are all correct for their positioning, but there is adequate citations in the locations you'd want to see them evidencing their arguments. The example presented to motivate the position/approach is compelling. The way the framework commits to inputs and outputs allows a good understanding without overcommitting to layer-internal design.

# Weaknesses

The paper could perhaps be a little sharper on its distinciton between the layers, to allow people who are less expert to understand the overall agentic framework under consideration. And whilst the example is compelling, the fact that the detection of the drifting behaviour requires comparisons across agent instances, but this kind of comparison is not explicitly accounted for in the system, makes it feel like the whole stack depends on something external to the position/proposed approach.


typo? ATLIS surfaces this drift before it causes harm at Layer 1 flagging a systematic

**Support:**

3

---

> ### Author Rebuttal · Authors · 2026-03-30
>
> We thank the reviewer for the thoughtful and encouraging assessment. We appreciate the recognition that the position is clearly articulated and balanced with alternative perspectives. ###
>
> ## Regarding W1: Distinction between ATLIS layers ##
>
> Our intent is that each layer corresponds to a different level of explanation and responsibility.
>
> To improve clarity between layers, we propose the following revision:
>
> Section 3.2 - Add a brief summary paragraph explicitly describing distinctions between layers and the type of questions each they answer, such as: “Layer 1 focuses on behavioural monitoring (*what is the system doing and is it behaving as expected?*); Layers 2–3 provide component and model-level interpretability (how did individual modules or agents reason, plan, and act?);  Layer 4 analyzes interaction and coordination across agents (*how did multi-agent interactions and execution trajectories lead to this outcome?*); and Layer 5 supports governance and oversight (*how should the system be monitored, audited, and adjusted across its lifecycle?*).”
>
> --------------------------
>
> ## Regarding W2: Comparisons across multiple agent instances ##
> We would like to acknowledge that ATLIS does support comparisons across agent instances. In practice, Layer 1 monitors patterns across multiple system runs, not just a single instance, allowing it to detect broader issues like behavioral drift or recurring coordination failures. Similarly, Layer 4 looks at how agents interact across different system runs, comparing decision paths and memory interaction patterns across to identify consistent deviations or failures.
>
> --------------------------
>
> #### Regarding W3: Typo at Line 360 (left)
> Fixed.
>
> --------------------------
>
> ## Regarding Q1: Comparisons across multiple agent instances ##
>
> ATLIS does support the comparisons across agent instances. In practice, Layer 1 monitors patterns across multiple system runs, not just a single instance, allowing it to detect broader issues like behavioural drift or recurring coordination failures. Similarly, Layer 4 looks at how agents interact across different system runs, comparing decision paths and memory interaction patterns across to identify consistent deviations or failures. This is also referenced in the response for Weakness #2 by reviewer.
>
> --------------------------
>
>  We appreciate the constructive feedback, and hope these edits strengthen unclarity and ratings around significance and discussion potential.

---

> > ### Author Rebuttal · Reviewer_e69w · 2026-04-01
> >
> > I think "multiple system runs" could still be clearer. My natural interpretation of that is multiple episodes of the same system instance, but if you mean something broader this should be clearer in the paper.
> >
> > I am happy with the wording for the layer distinctions - the purpose of each is now more grounded.

---

### Official Review · Reviewer_XcmD · 2026-03-12

**Significance:** 3
**Argument Clarity:** 3
**Rating:** 4
**Confidence:** 4

**Questions:**

1. Claim 1 (coevolution over reaction) is asserted with more confidence than it is defended. The response to Alternate View 1 address the concern that early interpretability methods may become obsolete as agentic architectures mature, but this argument is not entirely convincing. The paper does not engage with the possibility that coevolution could produce interpretability infrastructure tightly coupled to current architectures (e.g., transformer-based planning, chain-of-thought reasoning) in ways that would require costly further engineering as architectures shift. A more nuanced treatment of what aspects of ATLIS are architecture-agnostic versus architecture-specific would strengthen this claim.

2. The paper acknowledges (in response to Alternate View 3) that layers are not treated as isolated silos, but the treatment of cross-layer interactions remains underdeveloped. The ATLIS framework specifies what each layer analyzes and when it is activated, but does not specify how evidence from multiple layers is synthesized into a unified explanation. For instance, when Layer 1 flags a behavioral anomaly and Layer 4 identifies a delegation failure, what is the protocol for combining these signals? The "minimal sufficient depth" principle provides a termination criterion, but the paper would benefit from at least one example of cross-layer synthesis reasoning.

3. The two feedback loops (Figure 1) are described in terms of data flow, but the paper does not specify the decision criteria for when a loop is triggered. For instance, what threshold of behavioral discrepancy activates the Safety & Alignment Revision Loop versus the Monitoring Refinement Feedback Loop? Some discussion of escalation criteria would significantly improve operational clarity.

**Alternative Views Section:**

Yes

**Compliance With Llm Reviewing Policy A Conservative:**

Affirmed.

**Discussion Potential:**

3

**Final Justification:**

I keep my positive rating.

**Paper Summary:**

This position paper argues that the current paradigm of model-centric interpretability is fundamentally misaligned with the demands of deployed agentic AI systems. The authors make three claims: (1) interpretability methods must co-evolve with agentic capabilities rather than be retrofitted post-deployment; (2) agentic opacity is multi-layered and requires tailored analysis at each layer; and (3) interpretability must be integrated across the full deployment lifecycle rather than applied as a one-time audit.
To fufill these claims, the paper introduces ATLIS (Agentic Trajectory and Layered Interpretability Stack), a framework comprising five interpretability layers, i.e., Real-Time Behavioral Monitoring, Mechanistic Circuit Analysis, Abstraction-Level Bridging, Multi-Agent Analysis, and Safety and Alignment, mapped across a five-stage deployment lifecycle. The paper illustrates the framework with a healthcare referral case study and includes formal definitions grounding the framework in trajectory-based semantics.

**Position:**

Yes

**Position In Title:**

Yes

**Related Work:**

3

**Strengths And Weaknesses:**

The topic in this paper is very relevant to the agent era and it provide a comprehensive framework for characterizing the interpretability of the AI systems. The healthcare case study is concrete. For the weakness, please see the Question section.

**Support:**

3

---

> ### Author Rebuttal · Authors · 2026-03-30
>
> ### We thank the reviewer for the thoughtful feedback. Please see responses to weaknesses and questions to provide further clarification below. ###
>
> ## Regarding Q1: Claim of co-evolution over reaction ##
>
> We agree that the discussion would be strengthened with distinctions between architecture-agnostic and architecture-specific dependencies. Within ATLIS: Layer 2 (Mechanistic Circuit Analysis) is more tied to the internal structure of the model, Layer 3 (Abstraction Bridging)  is partly dependent on architecture, but not as much as Layer 2. Layers 1, 4, 5 are architecture agnostic, can be embedded early and remain as stable governance through architecture changes. This stratification strengthens the position’s case for co-evolution. As such the term here would imply iterative refinement of specific layers, as opposed to re-engineering the whole framework.
>
> To address this, we propose edits:
>
> L275 (Table 1) -  Add an additional column in Table 1 indicating Architectural Dependence of Layer 2 and 3
>
> L313 (right) - To briefly explain architecture-coupled dependencies for Layers 2-3 in Section 3.2 where methodological adaptations may be required.
>
> ## Regarding Q2: Synthesization of Layers Proposed in ATLIS ##
>
> We agree that information on cross-layer interactions can be strengthened. Our intended view is that ATLIS enables structured root-cause tracing across layers by integrating evidence from multiple levels of the system.
> We propose edits:
>
> L262 (left) - add in Section 3.2: “ATLIS supports step-by-step identification of the root cause across system layers, where signals from different layers are jointly considered to identify root causes. For example, a Layer 1 anomaly (e.g., unexpected model output) can be contextualized with Layer 3 signals (e.g., coordination or delegation patterns) and Layer 4 signals (e.g., system-level failures) to reconstruct a causal chain across the execution trajectory. The traversal terminates when a causal chain is established from the anomaly to a responsible component. “
>
> L375 (left) - include reference in Section 3.3 that the healthcare case study implicitly follows this traversal (L1 flags referral timing divergence → L2–L4 trace memory weighting and handoff dynamics → L5 evaluates safety bounds), and making the synthesis protocol explicit there will address this gap without requiring structural changes to the paper.
>
> ## Regarding Q3: Feedback Loops ##
>
> As a position paper, ATLIS deliberately does not prescribe fixed thresholds, as appropriate criteria will vary by domain and risk tolerance. However, to illustrate the intended operationalization: a Monitoring Refinement loop might be triggered by recurring anomaly flags across *N* consecutive monitoring windows without resolution, while a Safety & Alignment Revision loop would be triggered by incidents that propagate beyond Layer 4 containment or that involve safety constraint violations. We will clarify this distinction in the revised paper.
>
> --------------------------
>
> We appreciate the reviewer’s constructive comments, and hope these edits address any unclarity and strengthen ratings on the position’s significance and discussion potential.

---

> > ### Author Rebuttal · Reviewer_XcmD · 2026-04-02
> >
> > I would keep my positive rating.

---

### Official Review · Reviewer_yB9X · 2026-03-13

**Significance:** 4
**Argument Clarity:** 3
**Rating:** 5
**Confidence:** 4

**Questions:**

- The argument is that agentic interpretability is a systems problem and should be addressed as such. However, the building blocks for such systems that we have at hand, e.g., for interpreting just individual LLM outputs, are still poorly understand, and may potentially suffer from issues such as unidentifiability [1]. The question is: Does agentic system interpretability need effective model-level interpretability to be practically viable? Or is there a path to agentic interpretability that doesn't depend on solving that problem first?

## References

1. Méloux, Maxime, et al. "Everything, everywhere, all at once: is mechanistic interpretability identifiable?." arXiv preprint arXiv:2502.20914 (2025).

**Alternative Views Section:**

Yes

**Compliance With Llm Reviewing Policy A Conservative:**

Affirmed.

**Discussion Potential:**

3

**Final Justification:**

After taking into account the authors' rebuttal and promised edits to the final paper, I feel that the weaknesses of the paper will have been strengthened sufficiently to justify an increase in rating to 5 - Accept.

**Paper Summary:**

A position is taken that the field of AI interpretability needs to shift focus to the problem of interpretability for agentic systems, which the position argues is a distinct problem from explaining individual model outputs. The paper makes the following contributions: 1) defines "agentic system interpretability"--a system level analysis spanning behavior, interaction, and accountability--and 2) introduces a framework called ATLIS that tries to operationalize interpretability in agentic systems via: a) coevolution over reaction, b) layered decomposition, and c) lifecycle integration. A hypothetical healthcare case study is provided to illustrate ATLIS. The paper additionally suggests future research directions, discusses alternative views to the main position, and addresses assumptions and broader implications. In recognition of the broader challenges, a call is made for a coordinated effort across researchers, practitioners, regulators, and organizations to realize the goals of agentic system interpretability.

**Position:**

Yes

**Position In Title:**

Yes

**Related Work:**

2

**Strengths And Weaknesses:**

## Strengths

1. I found the position to be timely, relevant, and important to the ICML community.
2. Although I was confused by the precise arguments (see Weaknesses), at a high level I found the line of argumentation compelling; that is, model-level interpretability is not sufficient for agentic systems, and these techniques must be expanded to support multi-turn reasoning and planning and memory.
3. The ATLIS stack is interesting and must have taken some careful thought. I appreciate the perspective of interpretability for agentic systems as a systems engineering problem requiring a layered and lifecycle oriented approach.
4. I appreciated Section 4, which provided concrete future research suggestions and a discussion about the necessary coordination across research, engineering, policy, and governance.

## Weaknesses

1. The arguments in support of the position were lacking in clarity:

  1a. It wasn't very clear to me what the paper was trying to say about model-level interpretability methods. The paper jumps from saying that it is impossible to connect agent behavior to isolated model mechanisms (L64 left column and L68 right column), but then it seems that these techniques are critical to operationalizing the vision of agentic system interpretability, hence its use in Layers 2 and 3 of ATLIS.

  1b. It also wasn't very clear what the paper was trying to argue about co-evolution. This should be clearly defined and grounded in familiar terms. Claim 1 (pg. 2) defines this as co-development of agentic interpretability methodologies alongside agentic capability. Then coevolution is discussed in the specific context of systems design: "embedding interpretability requirements into each phase from pre-deployment onward..." (L186 pg 4). Then in ATLIS, coevolution is operationalized as feedback loops for adapting to incidents during deployment (pg 4).

2. Weak presentation of existing related work on interpretability for [multi-]agent systems. In Layer 4 (L315), where this is discussed, it was unclear if concepts such as "delegation-graph tracing", "shared-memory conflict detection", "protocol analysis", and "graph-based anomaly detection over interaction networks" were hypothetical or actual methods. The citation provided (Acharya et al., 2025) does not mention these. Layer 1 also seems to be founded on some single agent interpretability techniques. An organized related work section that situates "agentic system interpretability" in the context of this other work would be helpful.

**Support:**

3

---

> ### Author Rebuttal · Authors · 2026-03-30
>
> ### We thank the reviewer for the thoughtful feedback. Please see responses to weaknesses and questions below. ###
>
>
> ## Regarding W1.a: Role of model-level interpretability ##
> We agree with this. We would like to clarify the relationship between statements in L64–L68 and the use of model-level methods in ATLIS (Layers 2–3). Our intended position is that model-level interpretability exists but is not sufficient for agentic systems. The statements in L64–L68 were meant to emphasize a limitation of *completeness*, not applicability. ATLIS reflects this distinction: Layers 2–3 incorporate model-level interpretability to provide localized, component-level insight, while higher layers capture system-level properties to explain how behaviours emerge across components over time.
>
>
> We will revise L64–L68 to: “In an agentic system, the final decision may depend on how multiple components interact across time. Therefore, while model-level analysis may provide insight into individual steps, the overall behavior emerges from the interaction of memory updates, tool use, and coordination across components. Examining single model outputs in isolation would not be able to completely explain this.”
>
>
> ## Regarding W1.b: Defining co-evolution ##
> We agree the terminology could be refined to align with our intent. Our intended meaning is that co-evolution refers to the continuous, lifecycle-wide adaptation of interpretability mechanisms alongside agentic system behavior, design, deployment, and post-deployment monitoring. The instances highlighted by the reviewer reflect different lifecycle manifestations of the same principle: Claim 1 introduces co-evolution at a research level (interpretability advancements evolving with capability), L186 discusses it at a design level (embedding interpretability from pre-deployment), and ATLIS operationalizes it at a deployment level (feedback loops and monitoring).
>
> To resolve this ambiguity, we will revise the wording in L116 (Claim 1) to *co-design* rather than using *co-evolution* and introduce a single explicit definition early in the paper: “In this paper, we use *co-design* to imply intentional, simultaneous, mutually-informing development of two distinct subsystems, capabilities and interpretability infrastructure, rather than interpretability being added retrospectively after deployment.”
>
> --------------------------
>
>
> ## Regarding W2: Interpretability techniques related to agents ##
> The methods listed in Layer 4 (delegation-graph tracing, shared-memory conflict detection, protocol analysis, and graph-based anomaly detection) are intended as design-level specifications informed by the multi-agent systems literature, not citations to fully mature tools. Acharya et al. (2025) explored agentic architectural approaches, and we will enhance related works as per for the mentioned techniques with these revisions:
>
> L71 (left): Define “agent” and scope in the introduction with: “In the context of agentic systems, agent refers to LLM-powered entities that plan, use tools, and communicate via natural language (e.g., AutoGen, CrewAI, LangGraph-style architectures).”
>
> L256 (left): Replace with existing interpretability techniques for LLM-based multi-agent systems such as: “ interactive probing agent (Sheng et al. (2026), url: https://arxiv.org/html/2602.05446v1”).
>
> L322 (left): Rephrase to: “Recent simulations in multi-agent interaction traces use existing techniques that provide useful ways to understand the system’s behavior, in addition to model-internal methods (Tracing Coordination Dynamics in Multi-Turn LLM Discussions, by Parfenova et al. (2025), url: https://openreview.net/forum?id=YKdiS2oE3A#discussion).”
>
> --------------------------
>
> ## Regarding Q1: If agentic interpretability requires viable model-level interpretability ##
>
> Our position is that agentic system interpretability does not depend on fully solving model-level interpretability, nor does it require these methods to be fully reliable before system-level approaches can be useful. We agree that current model-level interpretability methods are useful but not fully applicable to agents, which is why we view model-level interpretability as providing partial, localized signals that can be incorporated into a broader system-level framework. We argue that while model-level interpretability gives us valuable insights, agentic system interpretability can be achieved through system-level interpretability mechanisms that do not depend on perfect model transparency.
>
> --------------------------
>
> We hope this response adequately addresses the reviewer's constructive comments. We believe the proposed revisions strengthen the paper's argument clarity, related work coverage, and discussion. We respectfully look forward to the reviewer's reconsideration of the score.

---

> > ### Author Rebuttal · Reviewer_yB9X · 2026-04-03
> >
> > * Regarding W1b: Thanks for the clarification. The suggested edits will improve the paper.
> > * Regarding W2: Thanks for clarifying. The suggested edits will improve the paper.
> >
> > Regarding W1a and Q1, I would say this is partially resolved. The techniques called out in Layer 3 Abstraction Bridging L299-314 are the "incomplete" model-level mech interp methods. Is there any prior work to point to that bridges "agentic system-level interpretability" with these mech interp (model-level) interpretability techniques?

---

### Decision · Program_Chairs · 2026-04-30

**Decision:**

Accept (regular)

**Comment:**

Reviewers agree with the need for a system-centric level of interpretability and they found the work to be important and timely. In addition, reviewers felt the authors did an excellent job thinking through the issues in the ATLIS framework and found the framework to be comprehensive and well-explained. Reviewers also appreciated the case study to help explain the approach. In terms of weaknesses, two reviewers found the section on co-evolution of interpretability methods to be confusing and lacking a depth of discussion. Reviewers also found the distinction between layers to be unclear and they felt the paper needed more discussion on how cross-layer interactions are handled. These weaknesses are minor relative to the strengths and overall, this is a strong position paper that will generate a good amount of discussion.